# The NHE3 Inhibitor Tenapanor Prevents Intestinal Obstructions in CFTR-Deleted Mice

**DOI:** 10.3390/ijms23179993

**Published:** 2022-09-01

**Authors:** Xinjie Tan, Archana Kini, Dorothee Römermann, Ursula Seidler

**Affiliations:** Department of Gastroenterology, Hannover Medical School, 30625 Hannover, Germany

**Keywords:** mucoviscidosis, DIOS, mucus, intestinal fluid absorption, sodium hydrogen exchange, chloride channel

## Abstract

Mutations in the CFTR chloride channel result in intestinal obstructive episodes in cystic fibrosis (CF) patients and in CF animal models. In this study, we explored the possibility of reducing the frequency of obstructive episodes in *cftr*^−/−^ mice through the oral application of a gut-selective NHE3 inhibitor tenapanor and searched for the underlying mechanisms involved. Sex- and age-matched *cftr*^+/+^ and *cftr*^−/−^ mice were orally gavaged twice daily with 30 mg kg^−1^ tenapanor or vehicle for a period of 21 days. Body weight and stool water content was assessed daily and gastrointestinal transit time (GTT) once weekly. The mice were sacrificed when an intestinal obstruction was suspected or after 21 days, and stool and tissues were collected for further analysis. Twenty-one day tenapanor application resulted in a significant increase in stool water content and stool alkalinity and a significant decrease in GTT in *cftr*^+/+^ and *cftr*^−/−^ mice. Tenapanor significantly reduced obstructive episodes to 8% compared to 46% in vehicle-treated *cftr*^−/−^ mice and prevented mucosal inflammation. A decrease in cryptal hyperproliferation, mucus accumulation, and mucosal mast cell number was also observed in tenapanor- compared to vehicle-treated, unobstructed *cftr*^−/−^ mice. Overall, oral tenapanor application prevented obstructive episodes in CFTR-deficient mice and was safe in *cftr*^+/+^ and *cftr*^−/−^ mice. These results suggest that tenapanor may be a safe and affordable adjunctive therapy in cystic fibrosis patients to alleviate constipation and prevent recurrent DIOS.

## 1. Introduction

Cystic fibrosis (CF) patients develop intestinal obstructions, called “meconium ileus” in newborns and “distal intestinal obstructive syndrome” (DIOS) in juveniles and adults. Radiologic signs and clinical symptoms are those of mechanical ileus. Surgery is to be avoided, if at all possible, and conservative treatment consists of orally and rectally applied Gastrografin (meglumine diatrizoate), polyethylene glycol (PEG) lavage (20%), and oral laxatives and enemas [1]. Nevertheless, the need for surgery is high, and the incidence of obstruction-related segmental colectomies is increasing, with high morbidity and mortality [2,3]. Recent Cochrane analyses revealed a clear lack of evidence for preventive, as well as therapeutic, treatment strategies and suggested that randomised controlled trials need to be conducted [4,5].

A recent survey revealed a high rate of abdominal complaints with a likely origin in the lower gut in CF patients not suffering from recurrent DIOS episodes [6]. Other studies demonstrated a correlation between abdominal symptoms and an increase in inflammatory markers in the stool [7,8], undermining the hypothesis derived from CF animal models that inflammation, secondary to bacterial overgrowth, predisposes to obstructive episodes, disease severity, and abdominal complications [9,10,11]. However, this hypothesis has not been validated in animal models or CF patients.

It has also been shown that the treatment with CF modulators and correctors reduces CF-associated inflammation [12]. Nevertheless, many of the patients who had undergone surgery for DIOS were already on CF corrector therapy, which is also not available for all CF mutations. This suggests that additional intestine-targeted therapies that improve the sequelae of CFTR dysfunction—namely, reduced gut fluidity, high luminal acidity, mucus hyper-viscosity, increased transit time, microbial dysbiosis, and mucosal inflammation—and, thereby, hopefully reduce obstructive episodes in the intestine are urgently needed. As it is obviously difficult to perform double-blind, placebo-controlled studies in this highly compromised group of patients, it is important to carefully test pharmacological strategies to alleviate CF-related intestinal disease in suitable animal models.

All currently available CF animal models display a very high incidence of intestinal obstruction, leading to death unless prevented by surgical, genetic, or pharmacological rescue [13,14,15,16]. *cftr*^−/−^ mice do not display pancreatic or pulmonary disease but consistently develop intestinal obstructive episodes. Hence, we considered this model ideal to test intestine-targeted drug therapies that have been approved for the treatment of constipation-prone irritable bowel disease and are believed to increase epithelial fluid secretion, decrease fluid absorption, or both.

In a previous report, we examined the efficacy of the guanylate cyclase 2C ligand linaclotide, the prostaglandin E1 analogue lubiprostone, and the intestine-specific Na^+^/H^+^ exchanger isoform 3 (NHE3) inhibitor tenapanor for jejunal and colonic fluid balance and alkaline output in both anesthetized *cftr*^−/−^ mice and F508del mutant mice in vivo, titrating each drug to the minimal concentration that elicited a significant increase in gut fluidity and alkalinity [17]. Although each of the drugs was able to increase jejunal alkaline output and reduce jejunal fluid absorption, linaclotide had no effect on the colon, and lubiprostone needed to be applied in much higher concentrations than were likely present in the gut lumen with the approved dosages for humans, suggesting that the anti-constipation effect may be due to an effect on motility rather than fluid balance. Tenapanor was able to decrease fluid absorption and increase alkaline output with similar efficacy in *cftr*^−/−^ and *cftr*^+/+^ mice and did so in concentrations that were reasonably close to what can be expected in the gut lumen with the approved dosage for humans. We therefore chose tenapanor to test its efficacy to prevent obstructive episodes in *cftr*^−/−^ mice when given by oral gavage for 21 days. We also aimed to explore the effect of tenapanor on the hallmarks of CF intestinal dysfunction; namely, reduced gut fluidity and alkalinity, mucus accumulation, cryptal hyperproliferation and villus elongation, slowed gastrointestinal transit time, and mucosal inflammation. 

## 2. Results

### 2.1. Increased Stool Water Content and More Alkaline Stool pH but No Difference in Body Weight between Vehicle- and Tenapanor-Treated Mice 

*cftr*^+/+^ and *cftr*^−/−^ mice were intragastrically gavaged with either vehicle (1x PBS, pH 6.0) or tenapanor (30 mg kg^−1^) twice a day for 21 days in the absence of oral PEG-containing laxative (Figure 1a). Weight and stool water content were recorded daily. Body weight was slightly higher in the *cftr*^+/+^ group due to a higher number of males (Figure 1b). It was stable during the treatment period during the 3 week period in the *cftr*^+/+^ group and slightly decreased in the *cftr*^−/−^ group (possibly due to the stress related to the gavage) but was not different between tenapanor- and vehicle-treated mice in either group (Figure 1c,d).

A significantly higher stool water content was observed in the tenapanor-treated *cftr*^+/+^ and *cftr*^−/−^ mice throughout the gavage period (Figure 1e,f). The vehicle-treated *cftr*^−/−^ mice displayed a decrease of stool water content post-removal of Oralav (Figure 1f). This was less prominent in the *cftr*^+/+^ mice (Figure 1e). In a subset of mice, we also measured the stool pH before and 3 h after gavage with tenapanor (Figure 1g) and found the pH to be significantly lower in *cftr*^−/−^ compared to *cftr*^+/+^ mice and to be significantly increased after tenapanor gavage.

### 2.2. Accelerated GI Transit Time upon Treatment with Tenapanor

Gastrointestinal transit time (GTT) is prolonged in CF patients and in CFTR-deficient mice [9,18,19,20]. We therefore measured GTT after oral gavage of 30 mg kg^−1^ tenapanor or vehicle once per week. Compared to *cftr*^+/+^, the *cftr*^−/−^ mice in the vehicle-treated group showed a significantly longer transit time (Figure 2a,c). Upon treatment with tenapanor, a significant decrease of transit time was observed from week 1 onwards in the *cftr*^−/−^ mice and in week 3 in the *cftr*^+/+^ mice (Figure 2a,c). This reduction of GTT in both vehicle-treated and tenapanor-treated *cftr*^+/+^ and *cftr*^−/−^ mice from week 1 to week 3 was independent of the effect of tenapanor on the stool water content in the 5 h during the determination of GTT, which showed a significant difference between vehicle- and tenapanor-treated mice of both genotypes (Figure 2b,d). Interestingly, a difference in the pre-gavage (0 h) stool water content (not significant until week 2 but significant in week 3) between the vehicle- and tenapanor-treated *cftr*^+/+^ mice but not in the *cftr*^−/−^ mice was observed (which would be a time point of approx. 11 h after their last gavage). This may indicate a different durability for tenapanor action in *cftr*^+/+^ and *cftr*^−/−^ intestine.

### 2.3. Significant Reduction in Obstructive Episodes from Tenapanor

Treatment with tenapanor resulted in a significantly lowered number of obstructive episodes in *cftr*^−/−^ mice (8.3% vs. 46.2%, *p* < 0.05) (Figure 3a) and a significantly better survival of 91.7% compared to 53.8% with vehicle treatment (because all mice with suspected obstruction were sacrificed). In the *cftr*^+/+^ group, survival was 100% both with tenapanor and with vehicle treatment. The site of obstruction was between the proximal and mid colon in most cases, and the segments in which the faecal material had accumulated were mostly restricted to the regions of the mid-distal ileum and the proximal colon (Figure 3b), as obstruction was suspected early. Table 1 shows the age and sex distribution of the obstructed and unobstructed *cftr*^−/−^ mice. While the total group of mice before gavage had near identical age and sex distributions in both the vehicle- and the tenapanor-treated groups, the mice that suffered an obstruction were predominately male and had young ages.

### 2.4. Tenapanor Treatment Reverses Cryptal Hyperproliferation in the Ileum and Proximal Colon

Assessment of the cryptal and villus length, the mucosal integrity, and the muscle layer thickness was performed in all mice of the treatment groups. In the ileum and proximal and mid-distal colon, vehicle-treated *cftr*^−/−^ mice displayed crypt villus/crypt elongation, as well as increased muscle layer thickness compared to the *cftr*^+/+^ mice in the ileum and colon but not the jejunum (Figure 4a–c). Treatment with tenapanor alleviated the crypt-villus/crypt elongation in the unobstructed ileum and proximal colon compared to the vehicle-treated *cftr*^−/−^ mice (Figure 4b, two middle panels). The increase in muscle layer thickness, seen in all segments except the jejunum in the *cftr*^−/−^ mice, was not significantly affected by tenapanor treatment. The jejunum did not display significant differences between the groups.

### 2.5. Obstructed Segments Display Distorted Architecture and Loss of Mucosal Integrity

The obstructed group of mice displayed a highly distorted epithelial architecture in the ileum and proximal colon, which were the segments that were regularly affected by the accumulation of food particles and faecal bacteria rich material (Figure 4a). The mucosal barrier was clearly destroyed and a clear separation between intra- and extracellular mucus was not possible in many instances, and areas were identified where the epithelium was sloughed off the lamina propria. All these changes occurred in the live mice within a relatively short time span and demonstrate the severe ulcerative behaviour of the luminal content in conjunction with the altered physiological state of the intestinal segment.

### 2.6. Tenapanor Treatment Results in Decreased Mucus Accumulation in cftr^−/−^ Intestine

Immunostaining against Muc2, the major component in intestinal secreted mucus, showed that tenapanor treatment reduced the total Muc2 mediated fluorescence intensity in relation to the Dapi-mediated fluorescence intensity compared to the vehicle treatment (Figure 5a,b). In order to determine whether the decrease was due to intracellular Muc2 in the goblet cell thecae or in the luminal (mucosa-attached plus secreted mucus), we optically separated the epithelium from the lumen in each section and quantified the two components (Appendix A). Due to the many different parameters that had to be studied in the intestine of the mice, including the mucosa-adherent microbiome (the results of which will be published by Soraya Shirazy-Beechey’s group), it was technically not possible to perform a Carnoy fixation that preserved the firmly adherent mucus layer. Therefore, the quantification of the extracellular mucus in the luminal compartment was only approximate. Nevertheless, the data show a significantly increased accumulation of mucus in the *cftr*^−/−^ mice (Figure 5a–c). Tenapanor significantly reduced the mucus accumulation in the ileum and proximal colon of the *cftr*^−/−^ mice compared to vehicle treatment, while it did not affect these parameters in *cftr*^+/+^ intestine (Figure 5a–d). In the segments proximal to the site of obstruction (ileum and proximal colon), mucus accumulation in the intracellular and luminal compartment was prominent, but the two components could often not be distinguished with certainty. *Muc2* mRNA expression was not different in the small intestine in either genotype or in the two treatment groups, but it was non-significantly reduced compared to *cftr*^+/+^ in the colon, similar to previous observations (Appendix A) [21]. In contrast, mRNA expression of the membrane-resident *Muc1* was increased in the *cftr*^−/−^ ileum and proximal colon, and tenapanor treatment reduced *Muc1* expression relative to vehicle treatment (Figure 5e).

### 2.7. Mast Cell Number Reduced in Tenapanor-Treated Intestine

An increase in innate immune cells, particularly mast cells, has been shown in the small intestine of *cftr*^−/−^ mice [22]. Importantly, a recent study highlighted that the mast cell-dependent immune function was associated with worse survival in F50del mutant mice [23]. We therefore studied mast cell numbers and the effect of tenapanor treatment. A significant increase of Toluidine Blue-stained mast cells was seen in all examined intestinal segments of the *cftr*^−/−^ compared to *cftr*^+/+^ mice (Figure 6a,b). Treatment with tenapanor resulted in a significant reduction in the number of mast cells in the jejunum and mid-distal colon of *cftr*^−/−^ mice, which were also the segments with the highest mast cell numbers in the epithelium (outside of Peyer’s patches). In other segments, while there was a similar trend, it did not reach statistical significance. No differences in mast cell number were seen between vehicle- and tenapanor-treated *cftr*^+/+^ mice groups (Figure 6a,b).

### 2.8. Leucocyte Infiltration in the Segments Proximal to the Site of Obstruction

Myeloperoxidase (MPO) staining was performed to assess infiltration with inflammatory cells. The MPO-positive cell count in the different segments of *cftr*^+/+^ and *cftr*^−/−^ intestine were very low, but there was a significant increase in the ileum and mid-distal colon between *cftr*^+/+^ and *cftr*^−/−^ mice, independent of treatment. Leucocyte infiltration was strongly increased in the segments immediately proximal to the site of obstructions but not in the mid-distal colon, which was the segment that was always distal to the obstruction (Figure 7a,b). To verify this finding, we measured the mRNA expression of *Ly6g*, a marker for differentiated neutrophils and granulocytes (Figure 7c). *Ly6g* expression was much higher in the segments proximal to the obstruction.

### 2.9. Strong Expression of Proinflammatory Cytokine Expression in the Intestinal Segments Proximal to the Obstruction

Proinflammatory cytokine expression is more sensitive for the detection of low level inflammation than MPO staining. *Tnfα*, *Mcp1*, and *Il1β* mRNA levels were assessed in the different intestinal segments of all gavaged mice (Figure 8a–c). The expression levels were low in *cftr*^+/+^ and *cftr*^−/−^ intestine, irrespective of treatment, except where an obstruction had occurred. Proinflammatory cytokine expression levels were significantly increased in all segments proximal to the obstructed site but not in the segment distal to the obstructed site (mid-distal colon). However, the relative increase was approx. tenfold higher in the ileum and proximal colon, the segments that were in contact with the faecal matter, than in the jejunum, which was usually free of faecal content when the obstruction was clinically detected and the mice sacrificed. Note the different y-axes for the individual bar graphs.

## 3. Discussion

The primary goal of the study was to assess the potential of the oral intestine-specific NHE3 inhibitor tenapanor for the prevention of intestinal obstruction in CFTR null mice. Several major obstacles were noticed during the study preparation and had to be addressed: tenapanor quickly precipitates from aqueous solutions at neutral or alkaline pH. Since microencapsulated tenapanor for use in mice is not available, it had to be applied by gavage, and dose-finding studies were necessary. We aimed at finding a tenapanor dose and application frequency that resulted in a stable but mild increase in stool water, because diarrhoea is quickly lethal for mice [24]. We found that 30 mg kg^−1^ tenapanor, applied twice daily by gavage, resulted in a mild increase in stool water, and it did not significantly influence the body weight of the mice. Surprisingly, the required dose was the same in *cftr*^−/−^ and *cftr*^+/+^ mice, suggesting the relative independence of the CFTR-mediated route of fluid secretion and the NHE3-mediated route of fluid absorption. Total or intestine-specific NHE3 deletion is known to result in hyponatremia, acidosis, and a decrease in blood pressure, resulting in massive hyperaldosteronism and strong upregulation of the expression of all ENaC (epithelial sodium channel) subunits, even in the more proximal colonic segments in which ENaC expression is usually very low [25,26]. As taking blood in quantities sufficient to determine serum electrolytes, acid/base parameters, and aldosterone levels was not feasible in this study, we therefore measured the mRNA expression of the three ENaC subunits in the mid-distal colon of the tenapanor-treated and vehicle-treated *cftr*^−/−^ and *cftr*^+/+^ mice (Appendix A), which is a very sensitive method to assess volume depletion. No difference in the mRNA expression for any of the subunits was found between tenapanor-treated and vehicle-treated *cftr*^+/+^ mice, and only a very small difference was observed in the expression of the *Enac* α-subunit between tenapanor-treated and vehicle-treated *cftr*^−/−^ mice. Thus, we believe that the tenapanor treatment did not result in grave systemic volume depletion. It is likely that the microencapsulation of tenapanor available for use in humans would permit lower doses in the CF patient population. An optimal dosing regimen to prevent obstructive episodes in CF patients without causing diarrhoea or other unwanted side effects needs to be established.

The *cftr*^−/−^ mice experienced a mild loss of body weight in both the tenapanor and vehicle group, which may have been related to an increased stress level due to the twice daily gavage. The lack of an effect of tenapanor on body weight is an important finding, because it is known that NHE3 function is required for PEPT1-mediated dipeptide absorption [27]. This is in agreement with a lack of a reduction in PEPT1-mediated substrate absorption during short-term tenapanor ingestion in healthy volunteers [28]. However, NHE3 has also been shown to be involved in the absorption of amino acids [29] and micronutrients [30]. Therefore, careful observation is necessary during long-term studies in CF patients.

Replacing the drinking fluid containing the osmotic laxative with tap water may result in an initiation of an obstructive episode within the first day (which becomes evident only several days later); we therefore started the tenapanor gavage on the last day of the oral laxative administration. It is well-known that the incidence of obstructive episodes (always lethal) in *cftr*^−/−^ mice is very high during weaning [31], with 90% of *cftr*^−/−^ mice dead by day 30 unless preventive measures are taken [14], and decreases in adulthood. Previous studies that aimed at pharmacological prevention of obstructive episodes in *cftr*^−/−^ mice have, therefore, administered the pharmaceutical agent to mice aged 3–6 weeks [32] and 20 days [33], respectively. For our study, the mice had to tolerate twice daily gavage and preferably not loose body weight due to this procedure; we therefore chose the age range of 9–13 weeks. This explains why the percentage of obstructions was lower in our study compared to the studies mentioned above.

The primary goal of our study was to answer the question of whether intestinal obstructive episodes can be prevented by tenapanor in a test group at high risk for obstructions, which was the CFTR null mouse (no CFTR protein present). Although our study was conducted in an age group in which the mice have a decreased risk compared to younger ages, the results showed that the risk was still high (46% within the observation time) and was reduced to 8% with tenapanor treatment.

Apart from obtaining a positive result in the study, we wanted to understand how tenapanor conveyed this reduction in obstructive episodes at a cellular and molecular level. As previously studied in intestinally perfused mice, the direct effect of tenapanor is a reduction in NHE3-mediated sodium and fluid absorption and an increase in the luminal alkalinity in both the small and large intestine [17], which was also seen in the stool of the mice in this study (Figure 1g). The normalization of the delayed GTT in the tenapanor-treated *cftr*^−/−^ mice is likely another key effect protecting the mice against obstructions.

Somewhat surprisingly, tenapanor treatment resulted in the reversal of other features of the intestinal abnormalities in *cftr*^−/−^ mice. Of particular interest to us was the finding that tenapanor treatment reversed the cryptal hyperproliferation in the ileum and proximal colon. A hyperproliferative response has been observed previously in *cftr*^−/−^ small [34] and large intestines [17]. The intracellular pH of *cftr*^−/−^-deficient enterocytes is more alkaline than that of corresponding *cftr*^+/+^ cells [35]. This feature is preserved in the crypts, including the stem cells of intestinal organoids [36], and is causally related to the increased proliferative rate, which is preserved in *cftr*^−/−^ intestinal organoids [37]. The presence of NHE3 has been reported both in undifferentiated and differentiated intestinal organoids [38]. Immunohistochemical staining of NHE3 overlaps with that of CFTR in the villus region and cryptal mouth region [39]. Therefore, it is possible that chronic tenapanor application reverses, in part, some of the pathological effects of the high intracellular pH on proliferative activity. A similar observation was made by Bradford et al. in their studies of mice that were homozygous negative for CFTR, for NHE3, or for both genes. While both *cftr*^−/−^ and *nhe3*^−/−^ mice displayed an increased cell proliferation in duodenal crypts, this increase was significantly reduced in the crypts of *cftr*^−/−^/*nhe3*^−/−^ mice [40]. Thus, long term tenapanor treatment, if proven safe in CF patients, may even have potential to lower the risk for GI malignancy.

Another striking observation in tenapanor-treated mice was a reduction in intestinal mucus accumulation. We previously reported an increase in the goblet cell counts, as well as enlarged goblet cell theca, in the mid-distal colon of *cftr*^−/−^ mice that did not display evidence of inflammation [17,21]. Due to the good documentation of crypt-villus hyperproliferation in *cftr*^−/−^ intestine [17,34], we used a fluorometric method to quantify the Muc2 immunofluorescence in relation to that of the Dapi-stained nuclei in this project. The total and extracellular Muc2/Dapi immunofluorescence was significantly reduced in tenapanor- vs. vehicle-treated mice in the ileum and proximal colon, while it was very high when an obstruction was present.

We also noted a shift in the expression of the *Muc* genes, with a non-significantly lower expression of *Muc2* mRNA and a significantly higher *Muc1* mRNA. The 21 day treatment with tenapanor reduced *Muc1* mRNA expression compared to vehicle treatment. The somewhat lower *Muc2* mRNA expression was also seen in the present study, with no effect from tenapanor. It has been shown that *Muc1* deletion, although not a major component of intestinal mucin, reduces the thickness of the firmly adherent mucus layer [41]. Muc1-deleted CF mice have reduced amounts of intestinal mucus and better survival compared to Muc1-expressing CF mice [42]. It is therefore likely that the observed tenapanor-induced changes in Muc1 expression and mucus accumulation play a major role in reducing obstructive episodes.

The intestinal tracts of patients with cystic fibrosis display an inflammatory phenotype in the absence of bacterial infection or enzyme-induced fibrosing colitis. Multiple explanations have been provided, including bacterial overgrowth due to dysmotility, oxidative stress, intracellular accumulation of mutated CFTR protein, and lack of pancreatic and/or intestinal defensins [43,44,45,46,47,48]. In mice, chronic exposure to PEG-containing laxative in the drinking fluid both increased survival and reduced the inflammatory signature in the intestine of these mice [14,40]. We previously reported no difference in proinflammatory cytokines in the different segments of *cftr*^−/−^ and *cftr*^+/+^ littermates while on the PEG-containing laxative, yet several features that had been correlated with an absent CFTR function (in addition to the reduced anion and fluid secretory response) were present: increased enterocyte pH_i_, an increased mucus content in the goblet cell theca, crypt-villus elongation, and decreased surface pH with fluid hyper-absorption [17,21]. In this study, a discreet but significant increase in the number of mast cells was observed in the lamina propria of the intestinal tract of vehicle-treated *cftr*^−/−^ mice compared to *cftr*^+/+^ mice, which was completely or partially prevented by tenapanor treatment. A very discreet increase in leucocyte cell infiltration was seen in the *cftr*^−/−^ ileum and mid-distal colon, not accompanied by an increase in any of the proinflammatory cytokines. In contrast, strong increases in leucocyte infiltration and in the expression of proinflammatory cytokines were only seen in mice with an obstruction and were restricted to the intestinal segments proximal to the obstruction that were filled with faecal matter. These segments also exhibited severe alterations in the mucosal architecture. The fact that the inflammatory changes were observed in the dilated and faecal matter-filled segments proximal to the site of obstruction, whereas the distal segments were not inflamed, suggests that the intestinal inflammation was a sequela of the obstruction and not its cause. 

A relative shortcoming in our study was that the application mode of the drug was relatively invasive and, therefore, the three-week study duration was fairly short. The number of mice permitted for the study by the animal committee was sufficient to reach the primary endpoint but was not enough to determine the significance for the reversal of all of the studied CF-associated intestinal abnormalities by tenapanor treatment in every intestinal segment. When we performed the GTT experiments, we noticed that the stool water content was still elevated in the *cftr*^+/+^ mice 11 h after the last dosing, but not in the *cftr*^−/−^ mice, suggesting that more frequent dosing, not feasible with this form of application, may have been optimal. Obviously, this problem will not exist in CF patient trials. For a tenapanor trial in CF patients, it will be necessary to define the patient population at high risk for intestinal complications and, preferably, test its prophylactic potential. Patients with meconium ileus at birth and prior DIOS episodes, as well as lung or liver transplanted patients, are at particularly high risk, and CF diabetes, pancreatic insufficiency, and female gender also carry a higher risk [1,49,50]. While corrector and potentiator therapy is beneficial for CFTR-dependent functional outcome parameters [51,52], adjunctive therapy specifically targeting the constipation and obstructive episodes appears necessary in many CF patient subgroups.

Tenapanor treatment is considered safe for humans [53,54]. It has not been tested in children so far, and the longest observation time in a published trial is 52 weeks. Given the good safety profile, clinical trials testing the effect of tenapanor on the intestinal symptoms of CF patients seem warranted and timely.

## 4. Materials and Methods

### 4.1. Animals and Experimental Protocol

All experiments were performed with wild-type mice (*cftr*^+/+^) and CFTR null mice (*cftr*^−/−^) (FVB/N-CFTRtm1CAM) [55]. Both *cftr*^+/+^ and *cftr*^−/−^ mice were bred and maintained at Hannover Medical School as previously described [56], except for the use of an energy-rich diet for transgene animals (1414MOD141003, Altromin, Lage, Germany) instead the low-fiber diet (C1013, Altromin, Lage, Germany) after the mice had survived the first 2 months of age. *cftr*^−/−^ and their respective wild-type littermates were co-housed (unless the males need to be separated due to fighting) and received identical diets (1414MOD141003, Altromin, Lage, Germany) and the osmotic laxative drinking solution (Oralav, 40 mmol L^−1^ Na_2_SO_4_, 75 mmol L^−1^ NaHCO_3_, 10 mmol L^−1^ NaCl, 10 KCl, 23 g L^−1^ PEG 4000) to prevent obstruction and enhance survival post-weaning.

The experimental protocol was performed in age-matched mice (9–13 weeks) with similar percentages of males and females. Post-withdrawal of Oralav, *cftr*^+/+^ and *cftr*^−/−^ mice were intragastrically gavaged twice daily (with a gap of 10–12 h) with either the vehicle (PBS, pH 6.0) or with tenapanor (30 mg kg^−1^) for 21 days. Mice were assigned into the vehicle- or tenapanor-treated groups randomly. The mice were monitored daily and sacrificed either at the end of the experiment or if obstruction was suspected. Mice were sacrificed when a suspicion of obstruction was present over two consecutive observation periods (one observation period every 12 h, lasting 12 h). Suspicion of obstruction was present when the mice had a prominent abdomen, hunched body composure, no passage of faeces during one 12 h observation period, or a mouse wellbeing score of 3 or higher. The severity and scoring was assessed based on a published severity assessment system that is recommended by our institute for animal research [57]. The abdomen was opened, the intestine was dissected and photographed, and the site of obstruction was verified. Post-sacrifice, the intestinal tissues were harvested for RT-PCR and histology.

The experimental protocol was approved by the Hannover Medical School Committee and an independent committee assembled by the local government. Animal studies are reported in accordance with the ARRIVE guidelines 2.0 [58]. The application and permission numbers are Az. 33.14-42502-04-14/1549 and Az. 33.12-42502-04-19/3197 for breeding “stressed strains” and Az. 33.9-42502-04-18/2829 for experimental procedures.

### 4.2. Whole GI Transit Time

The total GI transit time (GTT) using Carmine red-stained food pellets was measured once per week as previously described, with a few modifications [59]. Briefly, the mice were fasted overnight (approximately 8 h) and placed in a clean, transparent, empty cage prior to the first gavage of the day. They were then fed the red pellets. The total GI transit time was measured as the interval from the first food intake until the appearance of the first red stool pellet. 

### 4.3. Stool Water Content 

Stool water content was measured every morning prior to gavage as previously described [21]. Stool pellets were also collected every hour for 5 h during the GI transit time measurement.

### 4.4. Stool pH Measurement

Luminal stool pH was measured in *cftr*^+/+^ and *cftr*^−/−^ mice pre- and 3 h post-gavage with tenapanor. One to two fresh stool pellets were collected and placed in 50 μL unbuffered distilled water and the pH was determined with a pH-strip (Carl Roth, Karlsruhe, Germany). Following this, the pellets were homogenized and the pH was measured via a pre-calibrated TitraLab TIM 865 pH meter (Hach, Düsseldorf, Germany) with a Sension +5209 pH electrode (Hach, Düsseldorf, Germany).

### 4.5. Histology and Immunohistochemistry

Dissected intestinal segments (jejunum, ileum, colon) were fixed in 4% paraformaldehyde and embedded in paraffin according to standard procedures. Three-micrometer-thick sections were stained with hematoxylin and eosin (HE) and basic histological parameters, such as crypt-villus length, crypt depth, and muscle layer thickness, were assessed [60]. Crypt-villus length was defined as the distance from the crypt base to the apex of the villus in the jejunum and ileum. Crypt depth was measured from the basement of the crypt to the surface epithelium in the proximal and mid-distal colon. Three to eight villi or crypts were measured and averaged for every section. Muscle layer thickness was measured from serosa to inner muscularis mucosae. Intestinal sections were also stained with Toluidine Blue to detect mast cells numbers [22]. Images were acquired under a Leica DFC295 light microscope. Mast cells were then counted in five different sections per segment. The results are expressed as cells per mm^2^ (image field area).

The dissected intestinal segments were processed for immunohistochemical (IHC) staining as previously described [21]. The sections were incubated overnight at 4 °C with primary antibodies to Muc2 (1:200, Santa Cruz Biotechnology, Heidelberg, Germany, Cat#sc-15334, RRID: AB_2146667) and MPO (1:250, Proteintech, Planegg-Matinsried, Germany, Cat# 22225-1-AP, RRID:AB_2879037). They were then incubated with the corresponding secondary antibody (Goat anti-rabbit Alexa 568; 1:500, Thermofisher, Waltham, MA, USA, Cat #A-11036, RRID AB_10563566) and Dapi to detect the nuclei. The slides were mounted with Mowiol 4–88 and images were acquired on an Olympus FluoView™ FV1000 confocal microscope. Muc2 was quantified as previously described, with a few modifications [61]. Briefly, the intensity of green fluorescence indicating Muc2-stained mucins was quantified using Image J software. A region of interest (ROI) was determined (Appendix A) individually for intracellular and secreted mucins. Following the normalization of the detected area with the background, the integrated density (I.D) ratio of Muc2 to Dapi was calculated for total, as well as for the intracellular and secreted, mucins. The additive Muc2-to-Dapi ratio of the intracellular and secreted mucins represents the total Muc2. MPO-positive cells were counted by eye from images taken at 40× magnification in five different sections per segment. The results are expressed as cells per mm^2^ (image field area).

### 4.6. Quantitative PCR

Isolation of RNA, cDNA conversion, and quantitative real-time PCR from the various intestinal segments (jejunum, ileum, proximal colon, and mid-distal colon) were performed as per the manufacturer’s instructions and as previously described [17]. Ribosomal protein S9 (RPS9) was used as housekeeping gene. The list of primers used can be found in Appendix A. The total RNA was extracted with the RNeasy^®^ Mini Kit (Qiagen GmbH, Hilden, Germany) based on the manufacturer’s instructions. One microgram of RNA was reverse transcribed using the QuantiTect^®^ Reverse Transcription Kit (Qiagen GmbH, Hilden, Germany). cDNA was then diluted 1:20 with DNase-free water, and 4 μL of the dilution was used as a template for PCR. Each reaction contained 5 μL of qPCRBIO SyGreen Mix Lo-ROX (PCR Biosystems, Dueren, Germany), 4 µL of template, and an appropriate amount of primers.

### 4.7. Data and Statistical Analysis

The study design included a priori power analysis to determine the sample size for each experiment/subgroup. We used data from prior/similar experiments to determine the effect sizes and SDs. These were used in a power analysis with the software G * Power [62]. To determine the necessary sample sizes, we assumed a type I error of α = 5% and a statistical power of at least 80%. Mice were allocated to the experimental groups by a different person than the experimenter, and this person also regenotyped the mice after killing and allocated the genotype to the mouse number. A fully blinded experiment was not possible because the experienced experimenter had a high chance of correctly guessing the genotype based on the different phenotypes of knockout and wild-type mice. Sample sizes were calculated as biological replicates and were allocated equally to the experimental groups. Studies were designed to generate groups of equal size; however, accidental deaths (spontaneous or euthanasia due to disease) occurred during the preparatory period (transfer of the mice after group assignment to an animal experiment laboratory, one week obligatory acclimatization, removal of PEG laxative at day 1 of the experimental period), resulting in slightly uneven numbers in the *cftr*^−/−^ groups. The data were analysed with GraphPad Prism Version 8.0.2 (GraphPad Software Inc., San Diego, CA, USA). All results are presented as means ± SEM. The body weight, stool water content, and GI transit time were averaged for each time point. Unpaired Student’s *t*-tests (for parametric data with normal distribution) or non-parametric Mann–Whitney U tests were used for the comparisons between vehicle- and tenapanor-treated groups and/or within genotypes. The area under the curve was used to analyse the daily body weight and stool water content. Obstruction and survival curves were analysed with the Kaplan–Meier log-rank test. For multiple comparisons between the genotypes, the one-way ANOVA (for parametric data with normal distribution) was used with the Bonferroni post hoc analysis only if the F value of the ANOVA reached significance. If the normality test failed with one or both groups, the non-parametric Kruskal–Wallis test was performed. Significant differences are indicated as: # and * *p* < 0.05.

### 4.8. Materials

Tenapanor was purchased from Adooq Bioscience (Irvine, CA, USA). Carmine red was from SERVA Electrophoresis (Heidelberg, Germany). Mayer’s Hematoxylin Solution, Eosin Y Solution, and Toluidine Blue were all from Sigma-Aldrich (Deisenhofen, Germany). 

## Figures and Tables

**Figure 1 ijms-23-09993-f001:**
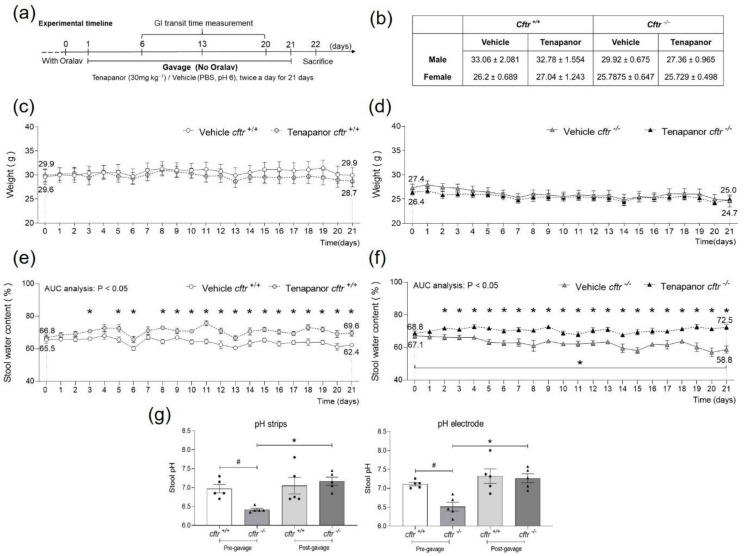
Effect of 3 week tenapanor treatment on body weight, stool water content, and stool pH in *cftr*^+/+^ and *cftr*^−/−^ mice. (**a**) Schematic description of experimental protocol. (**b**) Initial body weight (grams) of the male and female mice that did or did not develop obstructions. (**c**) The body weight remained stable during the 3 week gavage period in the *cftr*^+/+^ mice and (**d**) slightly decreased in the *cftr*^−/−^ mice, irrespective of treatment with tenapanor or vehicle. (**e**) A significantly higher stool water content was seen in the tenapanor-treated *cftr*^+/+^ and (**f**) *cftr*^−/−^ mice in comparison to the vehicle-treated groups. (**g**) Increased stool alkalinity 3 h post-gavage with tenapanor. Each dot/triangle represents one mouse. *cftr*^+/+^: n = 10 in both tenapanor-treated and vehicle-treated groups; *cftr*^−/−^: n = 12 in the tenapanor-treated and n = 13 in the vehicle-treated group. Data are represented as means ± SEM. # is used to represent the statistical analysis performed between the two different genotypes and * is used to represent the statistical analysis performed between the vehicle and tenapanor groups and between the same genotype, pre- and post-tenapanor gavage. # and *, *p* < 0.05.

**Figure 2 ijms-23-09993-f002:**
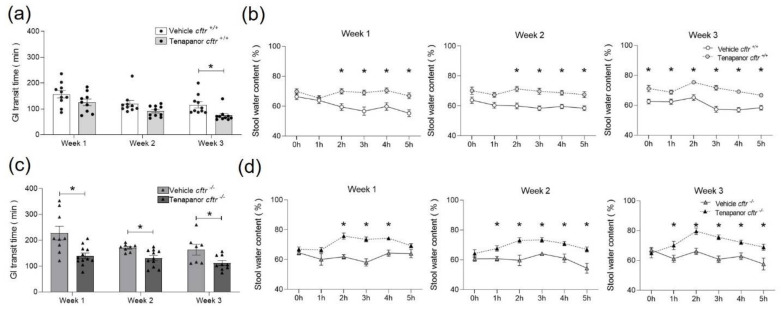
Faster GI transit time (GTT) in tenapanor-treated mice. (**a**,**c**) GTT was measured once per week, and it decreased in both *cftr*^+/+^ and *cftr*^−/−^ mice from week 1 to week 3 and was decreased after gavage with tenapanor compared to vehicle. (**b**,**d**) The longitudinal assessment of the stool water content during the 5 h post-gavage showed a significant increase with tenapanor compared to vehicle treatment. 0 h indicates stool samples collected pre-gavage. Each dot/triangle in the GI transit measurement represents one mouse, with the same mouse then being simultaneously assessed for stool water content. Data are represented as means ± SEM. * *p* < 0.05.

**Figure 3 ijms-23-09993-f003:**
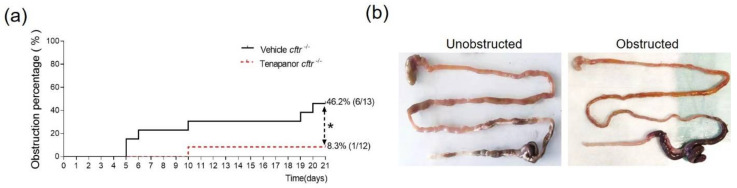
Significantly lower incidence of obstructions in tenapanor-treated *cftr*^−/−^ mice. (**a**) Treatment with tenapanor resulted in a significant reduction in the incidence of intestinal obstructions in *cftr*^−/−^ mice compared to the vehicle-treated *cftr*^−/−^ mice. (**b**) A representative image of the obstruction between the ileum and proximal colon is shown. All mice but one were suspected to have suffered an obstruction while alive and were sacrificed prior to death. * *p* < 0.05.

**Figure 4 ijms-23-09993-f004:**
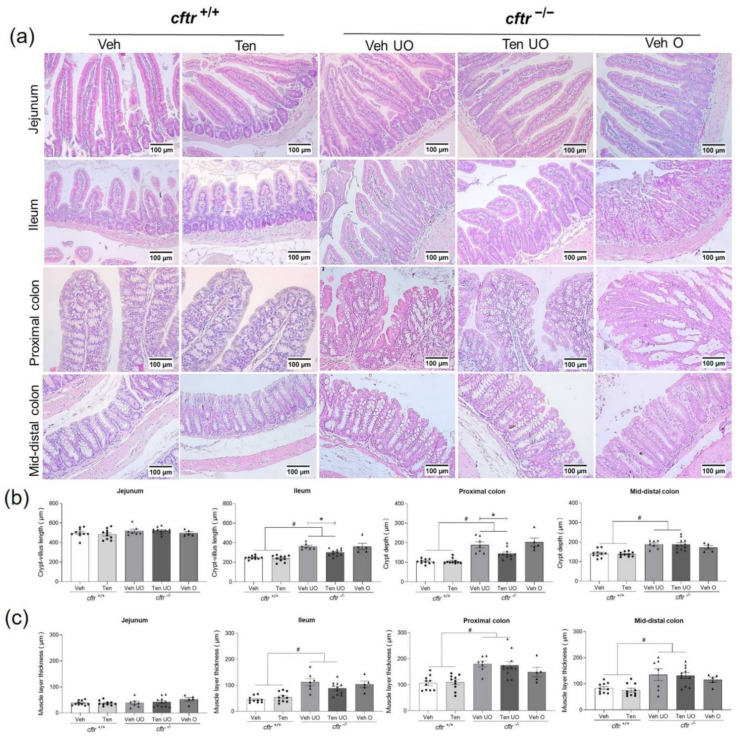
Reduced cryptal hyperproliferation in tenapanor-treated *cftr*^−/−^ mice. (**a**) Representative HE images of jejunum, ileum, proximal colon, and mid-distal colon in *cftr*^+/+^ and *cftr*^−/−^ mice treated either with vehicle or tenapanor. (**b**) Reduced cryptal-villus prolongation in ileum and reduced crypt prolongation in proximal colon in the tenapanor-treated *cftr*^−/−^ mice. (**c**) Muscle layer thickness displayed an increase in the ileum, proximal colon, and mid-distal colon of *cftr*^−/−^ mice, with no difference between vehicle- and tenapanor-treated groups. Each dot/triangle represents one mouse. For each mouse, five different (not serial) sections per segment were analysed and averaged. Veh—vehicle, Ten—tenapanor, UO—unobstructed, O—obstructed. Data are represented as means ± SEM. # is used to represent the statistical analysis performed between the two different genotypes and * is used to represent the statistical analysis performed between the vehicle and tenapanor groups. # and *, *p* < 0.05.

**Figure 5 ijms-23-09993-f005:**
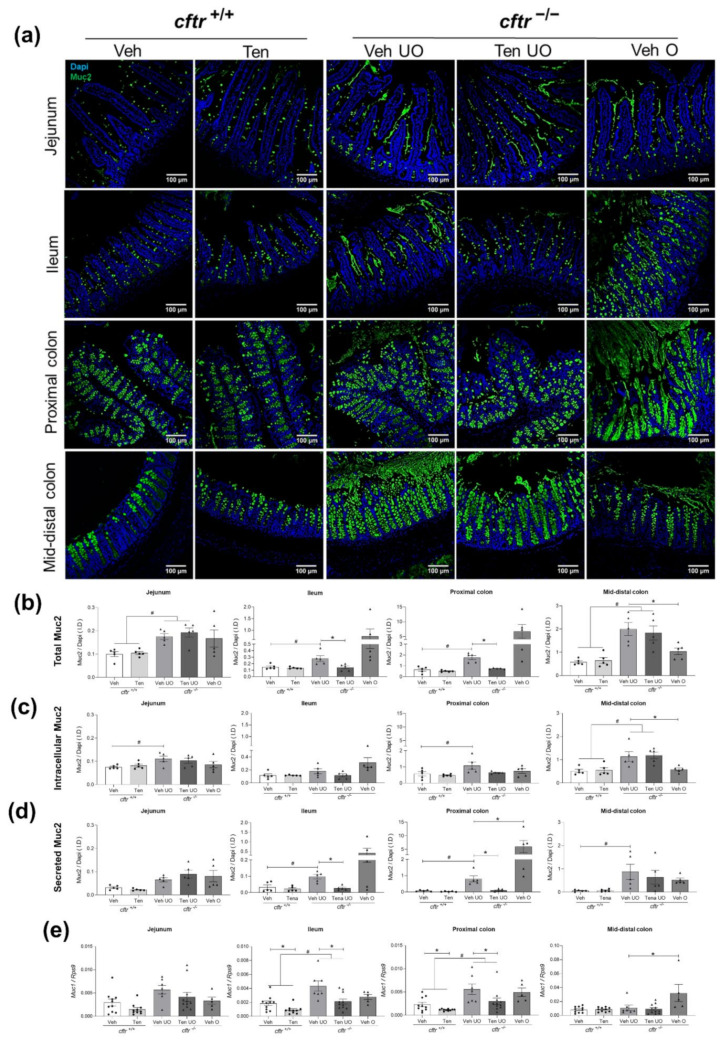
Decrease in mucus accumulation in tenapanor-treated *cftr*^−/−^ mice. (**a**) Representative images of Muc2 immunohistochemistry (major component of the secreted colonic mucin) of jejunum, ileum, proximal colon, and mid-distal colon in *cftr*^+/+^ and *cftr*^−/−^ mice treated with either vehicle or tenapanor (n = 5/genotype/group/intestinal segment). (**b**) Fluorescent intensity (I.D) ratio of Muc2 in relation to Dapi immunofluorescence (**c**) localized to the goblet cell thecae (intracellular mucus pool) and (**d**) localized to the luminal compartment (secreted mucus). (**e**) mRNA expression of *Muc1* in these intestinal segments. Each dot/triangle represents one mouse. Veh—vehicle, Ten—tenapanor, UO—unobstructed, O—obstructed, I.D—Integrated density. Data are represented as means ± SEM. # is used to represent the statistical analysis performed between the two different genotypes and * is used to represent the statistical analysis performed between the vehicle and tenapanor groups or between unobstructed and obstructed vehicle groups. # and *, *p* < 0.05.

**Figure 6 ijms-23-09993-f006:**
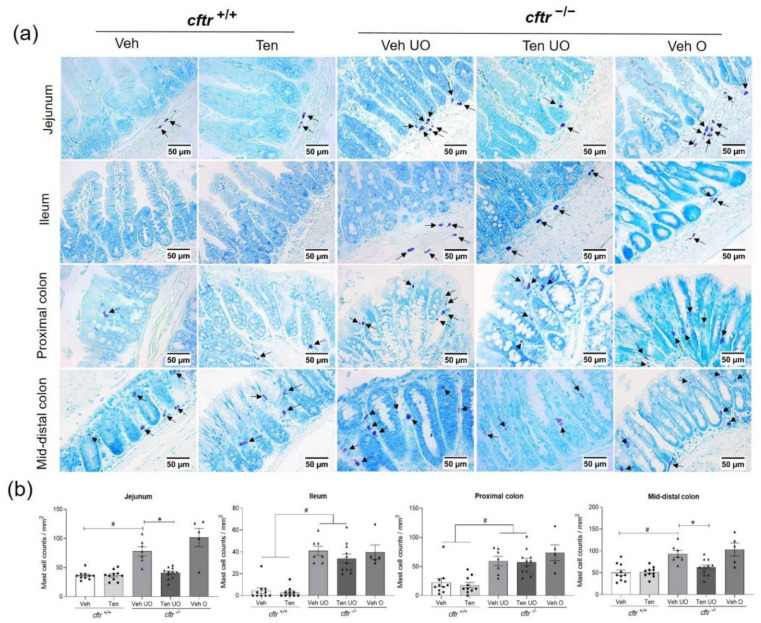
Reduced numbers of mast cells in tenapanor-treated *cftr*^−/−^ mice. (**a**) Representative images of Toluidine Blue-stained mast cells (indicated by black arrows) in the jejunum, ileum, proximal colon, and mid-distal colon in *cftr*^+/+^ and *cftr*^−/−^ mice treated with either the vehicle or tenapanor. (**b**) Quantified mast cell counts within the selected intestinal segments. Each dot/triangle represents one mouse. Veh—vehicle, Ten—tenapanor, UO—unobstructed, O—obstructed. Data are represented as means ± SEM. # is used to represent the statistical analysis performed between the two different genotypes and * is used to represent the statistical analysis performed between the vehicle and tenapanor groups. # and *, *p* < 0.05.

**Figure 7 ijms-23-09993-f007:**
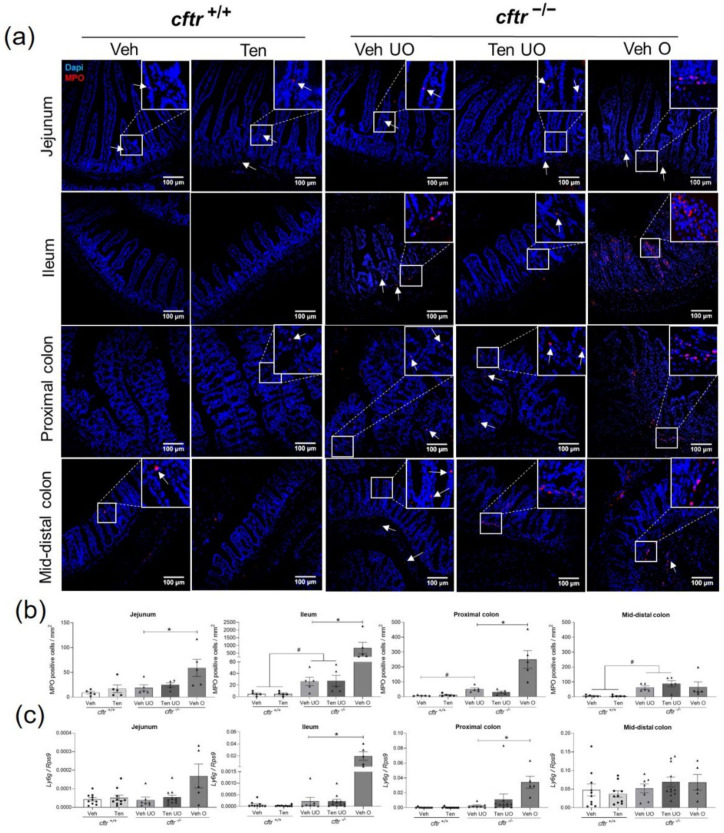
Leucocyte infiltration in obstructed *cftr*^−/−^ mice treated with vehicle. (**a**) Representative images of MPO-stained cells (indicated by white arrows) in the jejunum, ileum, proximal colon, and mid-distal colon in *cftr*^+/+^ and *cftr*^−/−^ mice treated with either the vehicle or tenapanor (n = 5/genotype/group/intestinal segment). (**b**) Counting of MPO-positive cells showed obvious leucocyte infiltration in the ileum and proximal colon of obstructed *cftr*^−/−^ mice treated with vehicle. (**c**) *Ly6g* mRNA expression further confirmed this observation. Each dot/triangle represents one mouse. Veh—vehicle, Ten—tenapanor, UO—unobstructed, O—obstructed. Data are represented as means ± SEM. # is used to represent the statistical analysis performed between the two different genotypes and * is used to represent the statistical analysis performed between the vehicle and tenapanor groups or between unobstructed and obstructed vehicle groups. # and *, *p* < 0.05.

**Figure 8 ijms-23-09993-f008:**
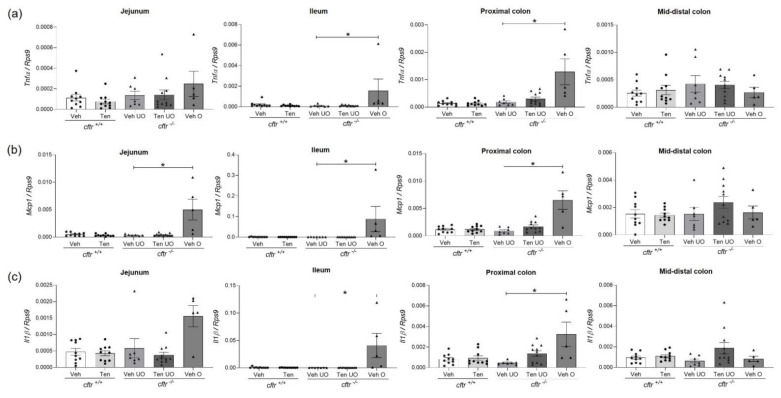
Inflammation in obstructed *cftr*^−/−^ mice treated with vehicle. The mRNA expression of pro-inflammatory cytokines (**a**) *Tnfα*, (**b**) *Mcp1*, and (**c**) *Il1β* were not significantly different between genotypes or between the tenapanor vs. vehicle-treated mice without obstruction but showed a dramatic increase in the ileum and proximal colon of obstructed *cftr*^−/−^ mice. Each dot/triangle represents one mouse. Veh—vehicle, Ten—tenapanor, UO—unobstructed, O—obstructed. Data are represented as means ± SEM. # is used to represent the statistical analysis performed between the two different genotypes and * is used to represent the statistical analysis performed between the vehicle and tenapanor groups or between unobstructed and obstructed vehicle groups. # and *, *p* < 0.05.

**Table 1 ijms-23-09993-t001:** Gender and age of the mice that developed obstruction and those that did not at the start of the experimental period.

	Gender				Age (Weeks)	
	Vehicle		Tenapanor		Vehicle	Tenapanor
	Male	Female	Male	Female		
**Unobstructed**	1	6	4	7	11.7 ± 0.522	11.5 ± 0.366
**Obstructed**	4	2	1	0	10.8 ± 0.477	9.0
**Total**	5	8	5	7	11.3 ± 0.365	11.3 ± 0.392

## Data Availability

The original data are available from the authors upon reasonable request.

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
