# Peer review of "The NHE3 Inhibitor Tenapanor Prevents Intestinal Obstructions in CFTR-Deleted Mice"

_ijms, 2022, doi:10.3390/ijms23179993_

Round 1

Reviewer 1 Report

The paper entitled “The NHE3 inhibitor tenapanor prevents intestinal obstructions in CFTR-deleted mice” describes the beneficial effects of oral treatment of tenapanor, a gastrointestinal NHE3 inhibitor, on intestinal obstructions in CFTR KO mice. The work is well-structured and properly conducted. However, I have a minor concern that need to be addressed before publication.

Intestinal epithelial cell-specific NHE3 knockout mice show a 25% mortality rate, metabolic acidosis, lower blood bicarbonate levels, hyponatremia and hyperkalemia associated with drastically elevated plasma aldosterone levels.

In this study, CFTR KO mice were treated with twice-daily dosing of tenapanor 30 mg/kg. Did the authors check the changes in blood pH, blood Na+ concentration, blood pressure, etc. in CFTR KO mice treated with tenapanor for 3 weeks?

Reviewer 2 Report

The primary goal of the study was to assess the potential of the oral intestine-specific NHE3 inhibitor tenapanor to prevent intestinal obstruction in CFTR null mice. 

This is a very interesting and original study on mice.

This study could suggest to start clinical trials testing the effect of tenapanor on the intestinal symptoms of CF patients.

I have no specific improvement to suggest to the autors

The conclusions are consistent with the argument presented and they address the main questione posed

The references are appropriate

I think that the tables and figure are ok
